# Identification of Preferential Recharge Zones in Karst Systems Based on the Correlation between the Spring Level and Precipitation: A Case Study from Jinan Spring Basin

**Yuan Chen** [1,2], **Longcang Shu** [1,2,*], **Hu Li** [3], **Portia Annabelle Opoku** [1,2], **Gang Li** [3], **Zexuan Xu** [4] and **Tiansong Qi** [1,2]

1    College of Hydrology and Water Resources, Hohai University, Nanjing 210098, China; chenyuan_916@163.com (Y.C.); portiafaith2015@gmail.com (P.A.O.); qitiansongg@163.com (T.Q.)
2    State Key Laboratory of Hydrology-Water Resources and Hydraulic Engineering, Hohai University, Nanjing 210098, China
3    Jinan Rail Transit Group Co., Ltd., Jinan 250101, China; lihu1007@163.com (H.L.); gang.li@live.cn (G.L.)
4    Climate and Ecosystem Sciences Division, Lawrence Berkeley National Laboratory, Berkeley, CA 94720, USA; zexuanxu@lbl.gov
*    Correspondence: lcshu@hhu.edu.cn; Tel.: +86-138-5194-1641

**Abstract:** The Jinan spring basin is located in the karst area of northern China, where springs serve as important sources of water supply. Several studies on spring protection and water supply have been carried out, and scholars have developed some laws on local groundwater flow dynamic and characteristics of aquifer structures. Unfortunately, there is a lack of detailed research on preferential recharge zones, which are the main recharge pathways of springs. Therefore, this research focuses on identifying preferential recharge zones based on the correlation between the spring level and precipitation. The results show that when precipitation is more intense or lasts longer, there is a stronger correlation between spring level and precipitation. It has been established that the precipitation at Donghongmiao station has the closest relationship with the dynamic of Baotu spring, which is found to be the most significant contribution to spring preservation. Two potential preferential recharge zones in the Jinan spring basin are detected through correlation analysis and geological exploration data. These findings support spring protection and water supply projects in karst regions.

**Keywords:** Jinan spring basin; spring level; rainfall; correlation; preferential recharge zone

## 1. Introduction

Several complex voids are created in aquifers as a result of the dissolution of carbonate rocks by water flow, such as fissures, conduits and caves, which are favorable for the storage and movement of freshwater, leading to the heterogeneity and vulnerability of karst aqueous mediums [1–4]. China has a massive karst region with complex karst landforms, and a quarter of its groundwater resources are distributed in karst areas, some of which typically emerge as springs [5]. The Jinan spring basin in northern China is a typical karst aqueous system, where springs serve as major discharge points of groundwater [6,7]. Springs are not only an indispensable source of water supply, but also help to preserve the water ecosystem of Spring City, Jinan. The continuous extraction of groundwater resources coupled with city construction and socio-economic development has destroyed the natural water balance of the Jinan karst aquifer system [8]. Records show that the spring flow has frequently diminished since the 1970s, wreaking havoc on the ecosystem environment as well as resident productivity and lifestyle. Spring protection and water supply engineering have been a crucial livelihood issue of great concerned to the government, and they have emerged as essential and demanding areas for hydrogeologists to investigate.

Karst aquifer systems have complex hydrogeological conditions with low-permeability pores and highly conductive conduits, and karst groundwater transport is a process of

groundwater pressure waves propagating through conduits, fractures and the matrix of karst aquifers [9,10]. Preferential recharge zones in karst water are master drainage networks primarily composed of conduit and fractured areas that locally modify the flow field in order to capture groundwater from the surrounding aquifer matrix, and are often created in certain geological structural belts, river valley belts or fissure concentrated zones [11,12]. In the Jinan spring basin, preferential recharge zones specify the dominant flow paths of aquifers and serve as the main spring supply channels. The location of preferential recharge zones must be investigated to protect the connectivity of preferential recharge zones, increase the performance of artificial recharge and optimize the groundwater monitoring network.

Preferential flow occurs in the vadose zone and saturated zone, but their methods of identification are quite different. In order to investigate the preferential flow paths in the shallow vadose zone, it is convenient to detect the dynamic change of soil water quality and quantity by installing probes such as various infiltrometers and lysimeters [13,14]. However, the composition of aquifers is complicated and monitoring wells and boreholes are unevenly distributed, rendering direct hydrogeological surveys for the study of preferential recharge zones time-consuming and difficult. Luo et al. [15] delineated two north–south preferential flow pathways and two stagnated locales based on the contours of groundwater age and spatial patterns of radionuclide disequilibria at INEEL (Idaho). Kishel and Gerla [16] used a spatial statistical study of measured hydraulic conductivity to infer the characteristics of preferential flow that groundwater discharges to lakes. Bolève et al. [17] and Robert et al. [18] demonstrated that Electrical Resistance Tomography (ERT) and Self-Potential (SP) methods could be used to identify fractured or karstified areas and detect flow paths. Eiche et al. [19] used hydrochemical and physico-chemical monitoring data to investigate the recharge source and flow movement of karst water in Gunung Kidul. In simple terms, previous research methods were all carried out in accordance with spatial heterogeneity of aquifer hydrologic characteristics and data analysis methods, but the acquisition of these data often necessitates a large amount of manpower and resources due to the inherent complexity of geologic structure of karst systems [20].

Karst area with thin surface soil layer usually has a slow rate of soil formation, high infiltration capacity and complex topography due to special geologic conditions except in arid and semi-arid areas [3,21]. As a result, precipitation penetrates heterogeneous karst aquifers rapidly and easily, causing the local groundwater level to rise, thereby affecting spring discharge. Therefore, it is feasible to study the characteristics of karst aquifer structures and change laws of groundwater resources based on the relationship between groundwater dynamics and precipitation [22]. Rahnemaei et al. [23] compared the physical characteristics of karst aquifers in the Maharlu basin using spectral analysis on daily data of precipitation and spring discharge. Dotsika et al. [24] investigated the isotopic composition of precipitation and spring water in Greece from the 1960s to the early 21st century, taking into account spatial heterogeneity. Filippini et al. [25] concluded on the evolvement rules of spring discharge and water quality in an alpine karst aquifer under various rainfall conditions. Yu et al. [26] used the gray correlation degree to validate and analyze the correlation between rainfall in the Jinan spring basin and the Baotu spring and Black Tiger spring levels, which were 0.858 and 0.859, respectively. Since groundwater pressure waves propagate through porous media, conduits and fractures and eventually emerge as springs, the dynamic change of springs can reflect the response process of regional groundwater system to precipitation. Precipitation data can be gathered from meteorological stations, and spring level and discharge readings can be obtained from self-monitoring points; such a data collection technique is simpler and more convenient than geological exploration. In sum, identifying the heterogeneity of karst systems by analyzing the correlation between spring dynamic and precipitation is reasonable and practical.

While precipitation infiltration is the most important recharge source of karst aquifers, and spring level fluctuation is highly dependent on the spatial-temporal distribution of precipitation, the hydrologic process by which precipitation transforms into karst water

and then emerges as springs is nonstationary and nonlinear [27]. To investigate the correlation between spring level and precipitation, statistical analysis techniques must be used, taking into account various time scales and the lag time between them [28]. The discrepancies in correlations between spring level and precipitations at different gauge stations demonstrate the heterogeneity of karst aquifers, which is one of the bases to identify the preferential recharge zones. A strong correlation indicates a high degree of karstification, which is advantageous for precipitation recharge and accelerates karst water drainage towards springs. Miao et al. [11] used the cross wavelet transform on the spring discharge of Niangziguan Springs and precipitations of neighboring counties to detect a strong karst groundwater runoff belt. An et al. [29] used precipitation and spring discharge data to identify groundwater quick flow belts in a karst spring catchment by the multi-taper method (MTM), a frequency spectrum analysis technique. Given the scarcity of hydrogeological data and the complexity of geological exploration, it is acceptable and needful to detect the preferential recharge zones in karst regions by evaluating the correlation between groundwater dynamic and precipitation.

This study selected precipitation data from various stations in the direct and indirect recharge area for the Jinan spring basin, and then used time series analysis methods to investigate the variation trends of spring level and precipitation. Correlation analysis between the spring level and precipitation can be performed after the selected data have been tested for representativeness and reliability. The results of correlation analysis and geological exploration data were combined to identify karst preferential recharge zones. The findings provide scientific evidence for underground space development and serve as critical reference points for spring protection and water supply engineering in the Jinan spring basin.

## 2. Study Area

The Jinan spring basin, with an area of 1794.58 km$^2$, is located in the central and western part of Shandong Province, between latitude of 35°59′–37°32′ N and longitude of 116°13′–117°58′ E, as shown in Figure 1. The basin is located at the intersection of low hills and an alluvial plain, with the southern part being rolling mountainous, the central part being low hilly land and the northern part consisting of a piedmont sloping plain and an alluvial plain. It also has two major rivers that flow from south to north, the Beidasha River and Yufu River. There are three major geological faults located in the basin, named Chaomidian Fault, Qianfoshan Fault and Wenhuaqiao Fault, which are permeable in the limestone formation according to geological data. The spring region has a warm temperate continental monsoon climate with a mean annual temperature of 14.2 °C and rainfall of 621.8 mm, and the amount of precipitation in the wet seasons (from June to September) accounts for about 75% of annual precipitation. Precipitation is distributed unevenly throughout the area, declining from southeast to northwest.

The Jinan spring basin, which includes four major springs (Figure 2), namely Baotu spring, Black Tiger spring, Pearl spring and Five Dragon Pool spring, is a flat monocline in general. Mashan Fault and Dongwu Fault are the eastern and western boundaries, respectively, based on the impermeability of faults. Carboniferous and Permian strata and some rock masses are the northern boundary, which belongs to flux boundary, and a surface watershed which is consistent with the underground is as the southern boundary. The paleo-metamorphic rock series of Mount Tai group of Archaeozoic formed the sedimentary basement of this area, and the cap strata from bottom to top successively belong to pre-Sinian, Cambrian, Ordovician and Quaternary, composing the area's spatial distribution characteristics of karst aquifers (Figures 3 and 4). Based on the analysis of borehole data and previous hydrogeological surveys, the geological body of Jinan spring basin can be divided into three layers: a phreatic aquifer, an aquitard, and a karst aquifer. Specifically, the third aquifer can be called an unconfined-confined karst aquifer, which corresponds to the Ordovician/Cambrian strata and has a uniform surface form. The major recharge sources of the karst system consist of precipitation infiltration recharge and concentrated

recharge of riverbed seepage (Table 1). The water budget information is constructed based on the hydrogeological data collected, which manifests that the amount of recharge is larger than that of discharge in 2012 and the positive discrepancy indicates an increase in aquifer storage. However, because precipitation patterns and human activities change each year, the table cannot represent a long-term steady-state condition. Precipitation in the indirect recharge area either infiltrates into karst aquifers through the Quaternary strata or reaches the direct recharge area as surface flow. Precipitation in the direct recharge area mainly percolates down the rock fissures and flows downstream into the karst system. The elevation of the discharge area is lower than that of the direct and indirect recharge area, and the water level of confined aquifers is higher than the elevation of the surface, which contributes to the groundwater discharge, including depression springs and ascending springs. In general, the karst water in the Jinan spring basin flows from southeast to northwest and eventually to the drainage area (Figure 1), and its overall movement direction is approximately analogous to the tilt direction of rock strata, but the contour map cannot show the local flow field, which is complicated and volatile.

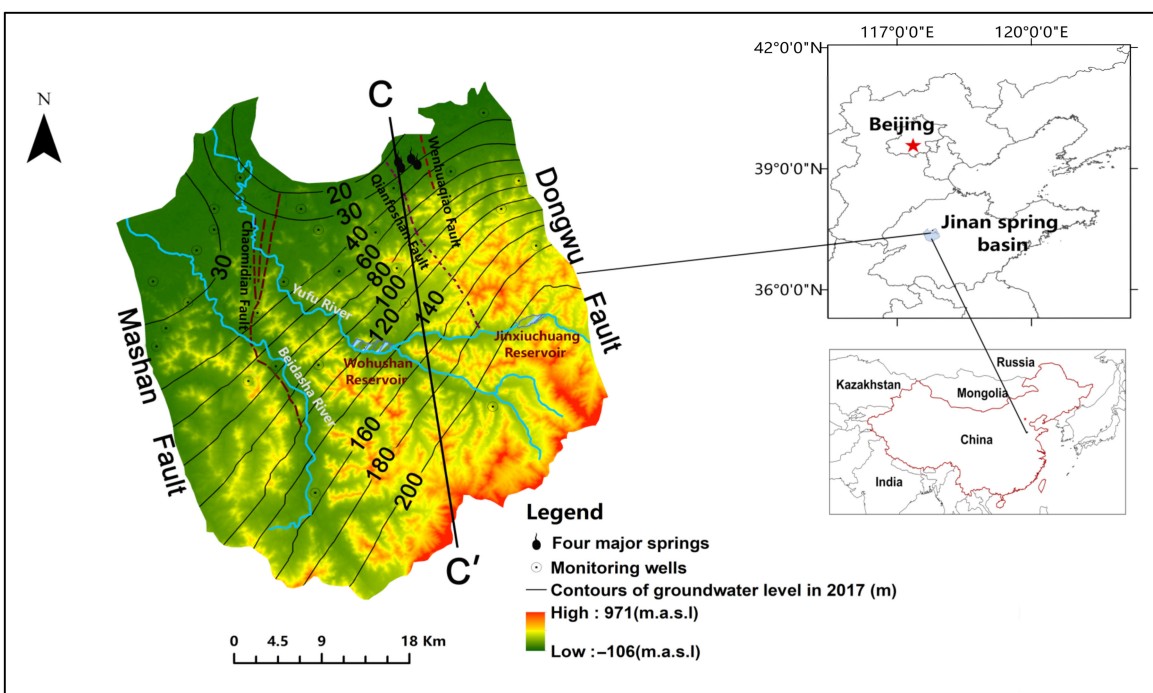

**Figure 1.** Topographic map of the Jinan spring basin in Shandong Province.

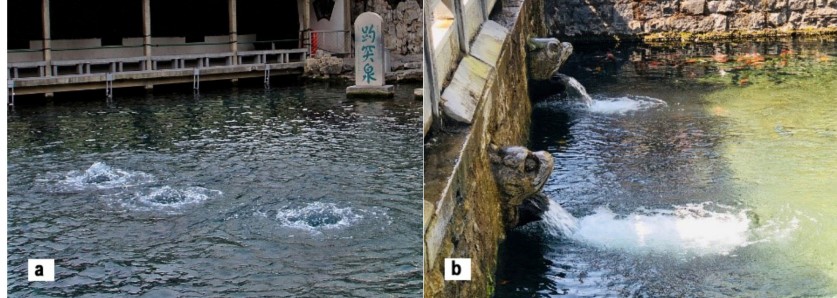

**Figure 2.** The conditions on site at the springs: (**a**) Baotu spring, (**b**) Heihu spring.

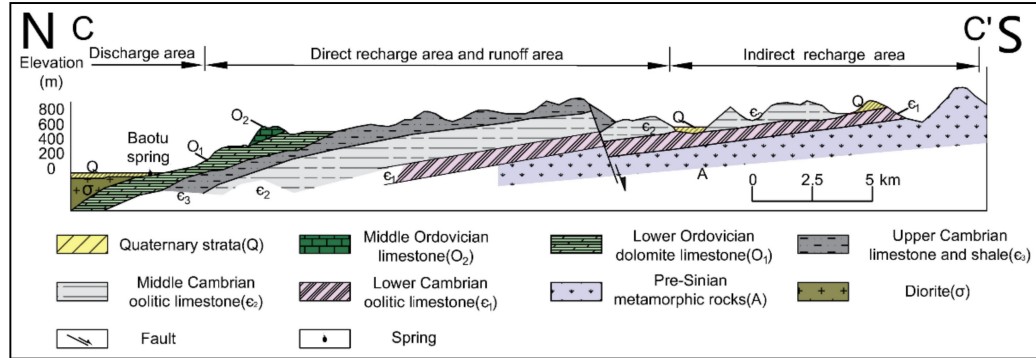

**Figure 3.** Profile map of Jinan spring basin in the north-south direction. The location of the cross-section CC′ is shown in Figure 1.

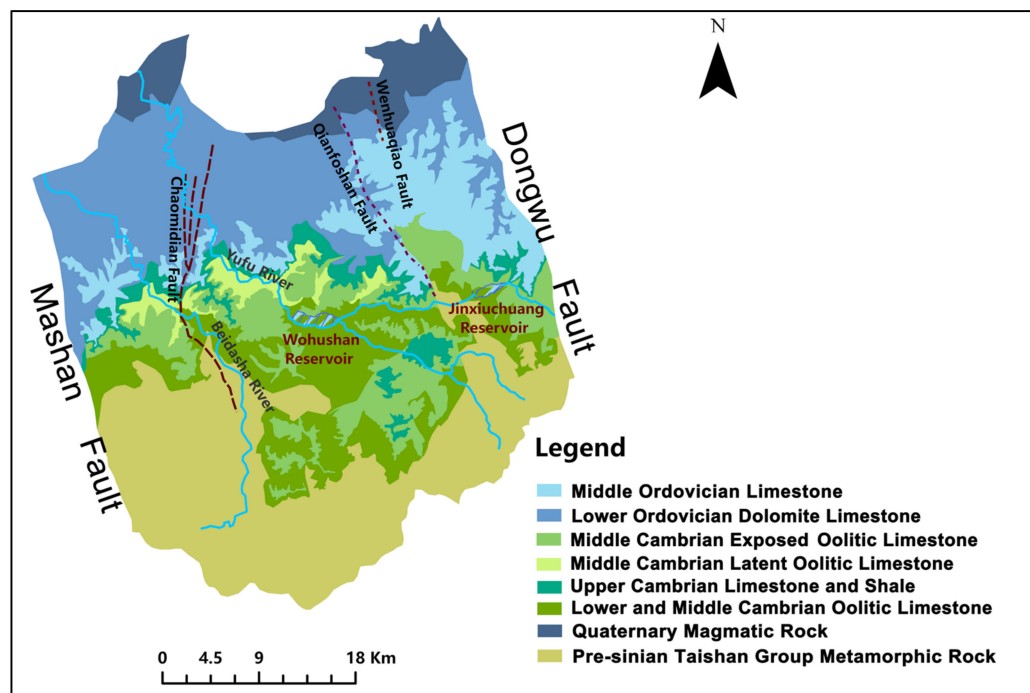

**Figure 4.** Hydrogeologic map of the Jinan spring basin.

**Table 1.** Groundwater balance of the Jinan spring basin for 2012.

|  | Item | Value (Million m³/d) | Percentage (%) |
|---|---|---|---|
| Recharge | Precipitation infiltration | 159.5 | 73.5 |
|  | River seepage | 26.36 | 12.1 |
|  | Irrigation seepage | 4.160 | 1.9 |
|  | Lateral recharge of groundwater | 18.16 | 8.4 |
|  | Artificial recharge | 8.800 | 4.1 |
|  | Total | 217.0 | 100 |

**Table 1.** *Cont.*

| | Item | Value (Million m³/d) | Percentage (%) |
|---|---|---|---|
| | Lateral discharge of groundwater | 9.950 | 5.7 |
| Discharge | Spring discharge | 69.04 | 39.9 |
| | Groundwater exploitation | 94.12 | 54.4 |
| | Total | 173.1 | 100 |
| | Imbalance | 43.87 | |

## 3. Materials and Methods

### 3.1. Data Sources

Daily precipitation data of 18 rainfall stations in the Jinan spring basin (Figure 5) were selected from the hydrological yearbooks of Hohai University from 2010 to 2016. We used the Thiessen polygon method to calculate the annual average precipitation, and the time range included wet years, normal years and dry years, making it suitable for statistical research. The Thiessen polygon method is relatively convenient and has previously been used to calculate the average precipitation for the Jinan spring basin [30]. The 18 rainfall stations are distributed over the direct recharge area, indirect recharge area and discharge area of the spring basin. Since the discharge area is downstream of the spring groups in the groundwater flow system, precipitations around the Changqing and Huangtaiqiao stations within the discharge area have a negligible impact on the dynamic of spring level, which are not included in this analysis. This study focuses on the remaining 16 precipitation stations that are evenly distributed in the southern mountains and piedmont plains, within the direct and indirect recharge region. Given that the recharge sources for the four major springs are comparable, the study focused on the Baotu and Black Tiger spring levels which are representative. The spring level data for Baotu spring and Black Tiger spring from 2012 to 2019 were obtained from Jinan Urban and Rural Water Authority's website. Daily groundwater level values of karst aquifers were measured by 28 monitoring wells (Figure 1), which could tap the limestone formation with the depths ranging from 80 m to more than 400 m. The borehole data and hydrogeological surveys manifest that the karst aquifer belongs to the Ordovician and Cambrian strata and has a uniform surface form. Groundwater level observation data can be used to calculate the hydraulic gradient and generate groundwater-level contour maps, which visually reflect the dynamic variation characteristics of regional groundwater flow systems, assisting in the determination of the orientation of preferential recharge zones to some extent. It is necessary to note that spring discharge data are usually used to perform relevant analyses, which greatly reflect the spring dynamic. In this study, we used the spring-level data because the spring level has a strong correlation with the spring discharge in the Jinan spring basin [31] and the data of spring discharge are not widely available.

### 3.2. Time Series Analysis Methods

It is important to research the representativeness of the data from selected years to evaluate the correlation between spring level and precipitation in detail. As a result, time series analysis was used to investigate variation trends and periods of spring level and precipitation data.

#### 3.2.1. Linear Trend Method

Let $x_i$ represent one variation with samples of size $n$ and $t_i$ serve as the corresponding time; then, the function of linear trend can be shown as $\hat{x}_i = a + bt_i$ $(i = 1, 2, \ldots, n)$, where $a$ and $b$ represent a constant and a gradient respectively, and $\hat{x}_i$ is the fitting value.

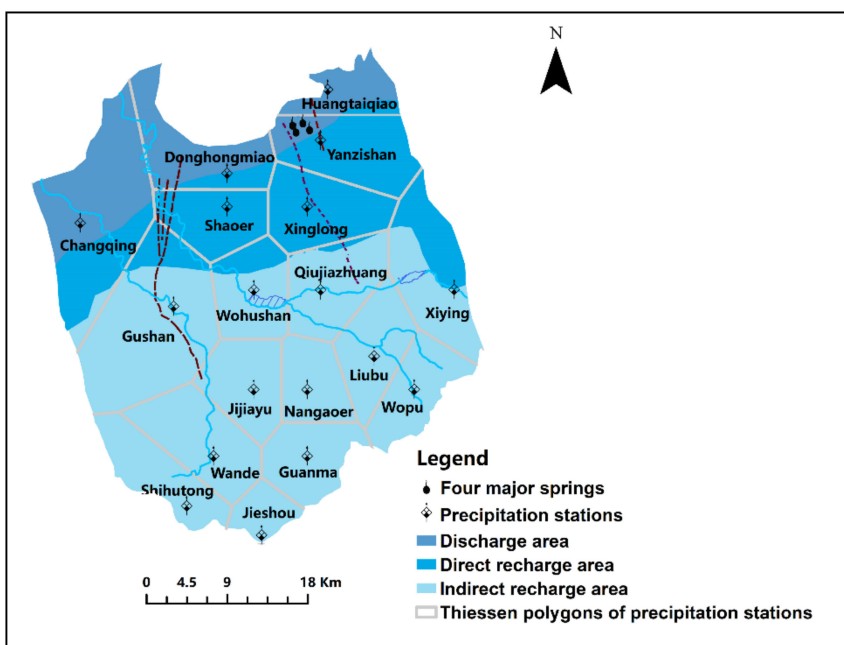

**Figure 5.** Distribution of precipitation stations in the Jinan spring basin and corresponding Thiessen polygons. See Figure 1 labels.

### 3.2.2. Moving Average Method

This method computes moving averages by sequentially adding new data and subtracting old data to eliminate contingent variables, whose function is analogous to a low pass filter to reveal variation trends based on the corrected smoothing arrays [32]. For the sequence $\{x(t)\}$ with samples of size $n$, series of moving average value $x_j$ can be expressed as:

$$x_j = \frac{1}{k}\sum_{i=1}^{k} x_{i+j-1} \ (j = 1, \ldots, n-k+1) \tag{1}$$

where $k$ is the time interval over which the average is calculated, $i$ represents a particular number from 1 to $k$ and $j$ is the revised sequence number.

### 3.2.3. Mann–Kendall Method

The M-K test is a non-parametric test method recommended by the World Meteorological Organization that is widely used around the world. A null hypothesis in this test is that the time series data $\{x(t)\}$ $(t = 1, 2, \ldots, n)$ are independent random samples with the same distribution [33,34].

The M-K trend test is based on the statistic $S$ defined as follows:

$$S = \sum_{i=1}^{n-1}\sum_{j=i+1}^{n} \text{Sgn}(x_j - x_i) \ (i, j \leq n \text{ and } i \neq j) \tag{2}$$

where $x_i$ and $x_j$ are the values in the time series $\{x(t)\}$ $(t = 1, 2, \ldots, n)$ and $\text{Sgn}(x_j - x_i)$ is a sign function as:

$$\text{Sgn}(x_j - x_i) = \begin{cases} +1, x_j - x_i > 0 \\ 0, x_j - x_i = 0 \\ -1, x_j - x_i < 0 \end{cases} \tag{3}$$

When the size of the time series ($n$) is greater than or equal to 10, the statistic $S$ approximately obeys the normal distribution. Its variance in terms of Var and the standard normal test statistic $Z$ are shown as:

$$\mathrm{Var}(S) = \frac{n(n-1)(2n+5)}{18} \tag{4}$$

$$Z = \begin{cases} \frac{S-1}{\sqrt{\mathrm{Var}(S)}}, & S > 0 \\ 0, & S = 0 \quad (n > 10) \\ \frac{S+1}{\sqrt{\mathrm{Var}(S)}}, & S < 0 \end{cases} \tag{5}$$

The testing of trends is performed at the significance level in terms of $\alpha$. If $|Z| \geq Z_{1-\alpha/2}$, the null hypothesis is rejected, and a significant trend exists in the time series. The value $Z$ implies an upward or downward tendency when it is greater or less than zero respectively. The value $Z_{1-\alpha/2}$ is obtained from the standard normal distribution table. When $\alpha$ is equal to 0.01 and the absolute value of $Z$ is bigger than $Z_{1-\alpha/2}$ (2.32), it represents passing 99% of the significance test.

The M-K mutation test is widely used to detect mutation points of hydrologic series and can determine when the mutation occurred. For the time series $\{x(t)\}$ with samples of size $n$, a new sequence $\{S_k\}$ is constructed:

$$S_k = \sum_{i=1}^{k} r_i \ (k = 2, 3, \dots, i \text{ and } i \leq n) \tag{6}$$

$$r_i = \begin{cases} +1, & x_i > x_j \\ 0, & x_i \leq x_j \end{cases} \ (j = 1, 2, \dots, i) \tag{7}$$

Under the hypothesis that time series data are independent random samples, a new statistic $UF_k$ is defined as:

$$UF_k = \frac{[S_k - \mathrm{E}(S_k)]}{\sqrt{\mathrm{Var}(S_k)}} \ (k = 1, 2, \dots, n) \tag{8}$$

where $UF_1$ is equal to zero, and $\mathrm{E}(S_k)$ and $\mathrm{Var}(S_k)$ are the mean and variance of the sequence $\{S_k\}$, respectively, shown as:

$$\begin{cases} \mathrm{E}(S_k) = \frac{k(k-1)}{4} \\ \mathrm{Var}(S_k) = \frac{k(k-1)(2k+5)}{72} \end{cases} \tag{9}$$

$UF_k$ is a statistical sequence calculated chronological by time series $\{x_1, x_2, \dots, x_n\}$. By repeating the process above in the opposite order $\{x_n, x_{n-1}, \dots, x_1\}$ and multiplying the results by $-1$, another sequence, $UB_k$, is obtained. The last step is plotting curves of these sequences ($UF_k$ and $UB_k$); if their intersection point is between the confidence lines, the time corresponding to the intersection is the beginning of mutation. When $\alpha$ is equal to 0.01 and the absolute value of $U$ is bigger than that of $U_{0.01}$ (2.58), it represents passing the significant test at level 0.01.

### 3.3. Correlation Analysis Statistical Metrics

The statistical software SPSS (Statistical Product and Service Solutions) was used in this study to investigate the correlation between the spring level and precipitation. The correlation analysis between spring level and precipitation was more credible and accurate after evaluating and comparing calculation results from different methods. The preferential recharge zones in karst aquifers of the Jinan spring basin were determined on this basis.

### 3.3.1. Pearson Correlation Coefficient Method

Pearson correlation coefficients are acquired by estimating the covariance and standard deviation of two variables, $x_i$ and $y_i$, from different sample series, defined as [35]:

$$R = \frac{\sum\limits_{i=1}^{n}(x_i - x)(y_i - y)}{\sqrt{\sum\limits_{i=1}^{n}(x_i - x)^2}\sqrt{\sum\limits_{i=1}^{n}(y_i - y)^2}} \quad (i = 1, 2, \ldots, n) \tag{10}$$

The value of $R$ varies from $-1$ to 1, with zero indicating that there is no correlation between the two sequences.

### 3.3.2. Cross-Correlation Analysis Method

This method is commonly used to analyze time series data for determining the greater correlation coefficient between two sets of data using dislocation movement based on the lag time. For two time series with the same length $N$, the statistic of cross-correlation is given by [36]:

$$Q_{CC}(n) = N^2 \sum_{t=1}^{n} \frac{C_t^2}{N - t} \tag{11}$$

where $C_t$ is defined as:

$$C_t = \frac{\sum\limits_{k=t+1}^{N} x(k)y(k - t)}{\sqrt{\sum\limits_{k=1}^{N} x(k)^2 \sum\limits_{k=1}^{N} y(k)^2}} \tag{12}$$

The statistic of cross-correlation approximates to $\chi^2(n)$ distribution with $n$ degrees of freedom. The method can be used to test the null hypothesis, which states that none of the first $n$ cross-correlation coefficients are different from zero. If the cross-correlation statistic surpasses the threshold of $\chi^2$ distribution, the dependency between these time series is statistically significant, and vice versa.

### 3.3.3. Multiple Linear Regression Method

Linear regression is a statistical analysis method that uses formulas to determine quantitative dependencies among two or more variables. Taking the bivariate linear regression model as an example, the equation is $\hat{y}_i = b_0 + b_1 x_1 + b_2 x_2 + \mu_i$, where $b_0$, $b_1$ and $b_2$ are partial regression coefficients and $\mu_i$ represents its random error term [37].

We used the selected data to evaluate the correlation between the spring level and precipitation based on the correlation analysis statistical metrics after passing a series of empirical tests including trends and periods derived from time analysis methods. The Pearson correlation coefficient method was used with daily data of spring level and precipitation from 2012 to 2016, and the cross-correlation analysis method was used with daily data in the wet seasons from June to September in 2012–2016, as was the multiple linear regression method.

## 4. Results

### 4.1. Variation Trends of the Spring Level and Precipitation Process

The hydrographs of Baotu spring and Black Tiger spring from 2012 to 2019 (Figure 6a) are nearly identical, with no noticeable difference despite the 1.5 km gap between them. Due to the similarity between their hydrographs with a Pearson correlation coefficient edging to one, the research work mainly focused on the Baotu spring. The red warning level of Baotu spring is 27.6 m, which is set to warn people of the spring cutoff probability. When the spring level falls below the red warning level, relevant agencies must initiate a first-level contingency plan to recharge groundwater for spring protection. As shown in Figure 6a, the period when the spring level of the Baotu spring exceeds its red warning level accounts for 93% of the time span and the gradient of its linear trend is $-0.0003$. The M-K trend test value Z of the monthly average spring level is $-4.32$, whose absolute value is greater than $Z_{1-\alpha/2}$ ($\alpha = 0.01$), indicating a statistically significant downward trend.

These two points of intersections of curves UF and UB between September 2013 and July 2014 are within the confidence intervals (Figure 6b), which are likely mutation points, indicating that the fluctuation of spring level remains slightly unstable. Since 2013 is a normal year and 2014 is a dry year, the mutation phenomenon is most likely triggered by a shift in precipitation pattern and an increased requirement for groundwater extraction.

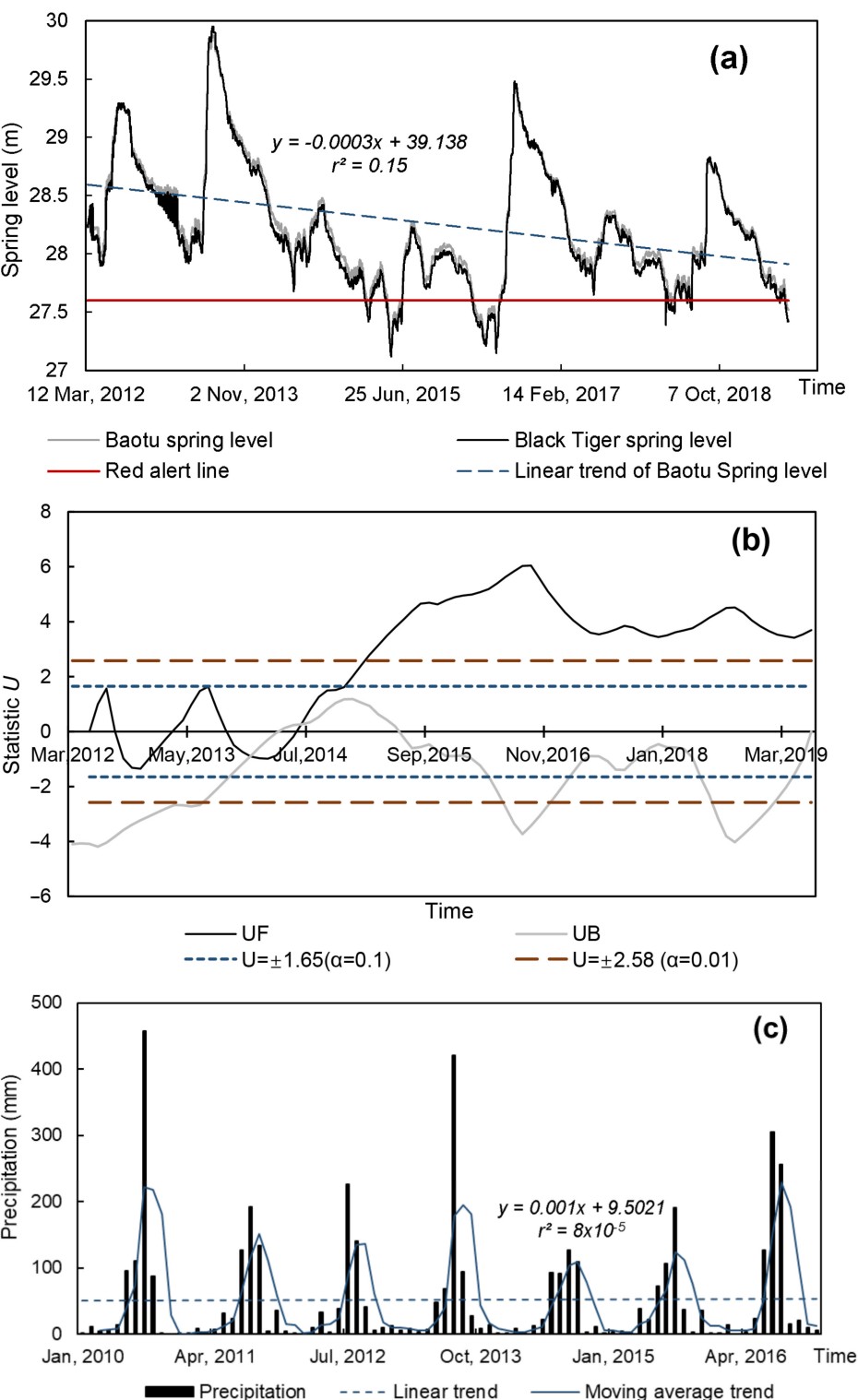

**Figure 6.** *Cont.*

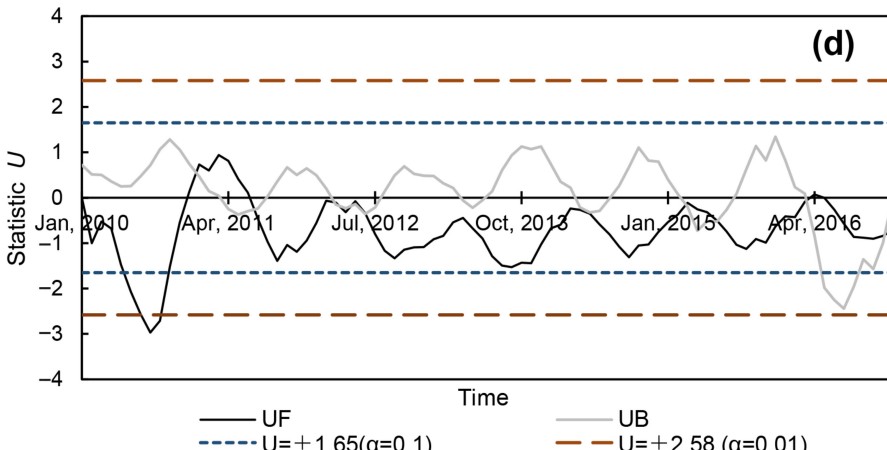

**Figure 6.** (**a**) Daily water level hydrographs of Baotu spring and Black Tiger spring with the linear trend of Baotu spring level, (**b**) statistic curves of monthly mean water level of Baotu spring calculated by M-K mutation test, (**c**) monthly mean precipitation hydrograph of Jinan spring basin with its linear trend and moving average trend, (**d**) statistic curves of monthly mean precipitation calculated by M-K mutation test.

For the monthly mean precipitation in the Jinan spring basin from 2010 to 2016 (Figure 6c), the gradient of the linear trend is 0.001, revealing a gentle variation in the precipitation process over seven years. Furthermore, the three-month moving average curve (Figure 6c) illustrates a continuous fluctuation with a series of wet seasons and dry seasons on an annual scale. The M-K trend test value $Z$ (0.71) of regional average precipitation is smaller than $Z_{1-\alpha/2}$ ($\alpha$ = 0.01), which reveals a statistically insignificant upward trend. The M-K mutation statistical curves in monthly average precipitation (Figure 6d) indicate that the UF and UB curves are approximately within the confidence intervals, showing a series of small amplitudes.

Natural and anthropogenic changes in hydro-climatic conditions are the major contributors to changes in spring dynamic [38,39]. The above statistical analysis shows a weak downward trend in spring level under a steady precipitation condition, which may be attributed to the effect of human activities on the springs over the years, such as groundwater exploitation and changes in land use types, especially in the recharge area. The purpose of analyzing the variation trends of the spring level and precipitation process is to verify the representativeness of the data used. These findings suggest that the changes in the time series are generally stable, and that the dataset chosen is appropriate for this study. Given that eliminating the periodicity and trend from time series is usually performed based on several decades of data [40,41], and that the periodicities of these time series are not distinct and there is much uncertainty regarding detrending, such work was not carried out in this study.

### 4.2. Characteristics of the Correlation Coefficients and Lag Response Time

The statistical approach we used to analyze the relationship between the spring level and precipitation in this section is the Pearson correlation coefficient method. Several studies have shown that the lag response time of spring level to precipitation is no more than one year [29,39,42]. As a result, the time series of spring level was manually moved back to make the spring level sequence lag behind that of precipitation, with different time intervals ranging from one month to twelve months, respectively, in which case the correlation analysis with time lags could be performed.

Using the 2014 and 2016 findings as examples (Figure 7), there is consistency in the correlation coefficient curves between the spring level and precipitations of different stations, and the peak points correspond to the same lag time, although the maximum distance between any two of these 16 stations chosen is up to 40 km. This means that

the distance between the stations and the springs and the hydrogeological conditions of stations have no significant relationships with lag time on a monthly scale.

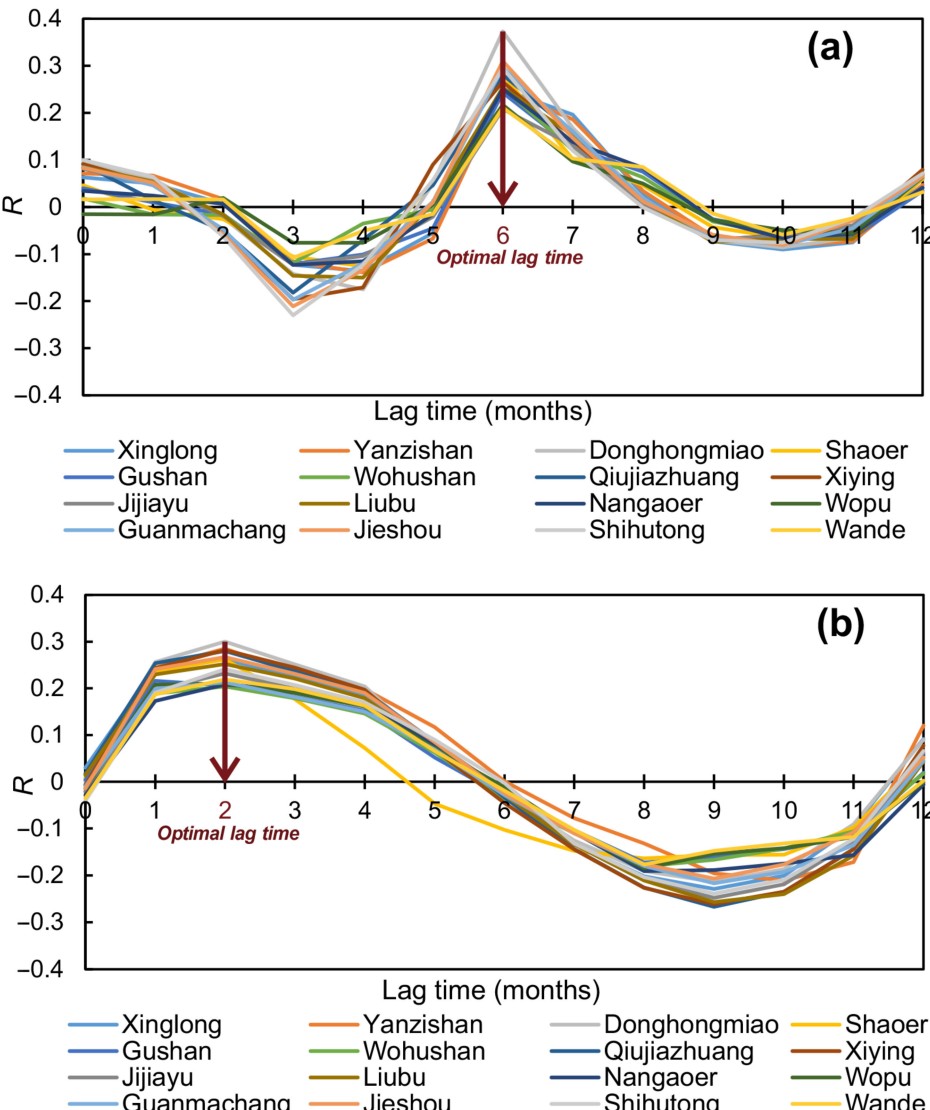

**Figure 7.** The change processes of Pearson correlation coefficient (*R*) with lag response time between the Baotu spring level and precipitations of 16 stations based on daily data in 2014 (**a**) and 2016 (**b**).

Furthermore, the optimum lag time corresponding to the highest Pearson correlation coefficient, namely the peak of the curve for each year, varies due to the annual precipitation variability and the impact of human activities. Linear trend analysis between the optimum lag response time and annual mean rainfall supports the notion that they are negatively correlated. As a result, it is probable that the heavy precipitation accelerates infiltration and shortens the time it takes groundwater to reach springs in karst regions. Furthermore, as the annual precipitation declines, the year's highest correlation coefficient between the spring level and precipitation indicates a downward trend based on the linear trend analysis. This may be due to the continuous low intense rainfall events in dry years, which results in a limited amount of infiltration, and consequently makes it impossible for groundwater flow to travel to the discharge area intensively. Another reason is that the low annual precipitation during dry years increases groundwater exploitation by some industries, especially agricultural industries, which rely heavily on groundwater resources.

### 4.3. Correlation between the Spring Level and Precipitations of Different Stations

This section focuses on the precipitation stations located in the recharge area, where precipitation has a strong influence on the dynamic of springs, and can serve as evidence for the subsequent analysis of preferential recharge zones in karst water. The highest Pearson correlation coefficient of each year corresponds to an optimum lag response time, based on which annual time series of spring level must be delayed relative to the sequence of precipitation. The Pearson correlation coefficient method can then be used to evaluate the correlation between the spring level and precipitation at different scales of cumulative precipitation, including one day, five days and ten days (Table 2). It appears that the correlation coefficient between them increases with cumulative time scale, which shows that aggregate precipitation over a long period can better reflect the process of precipitation infiltrating into groundwater and subsequently moving to the discharge area in concentration. However, if the aggregation periodicity is too long, the correlation between the extreme points will be weakened; therefore, the analysis in the next section is based on a one-day scale with heavy rainfall stages.

**Table 2.** The Pearson correlation coefficients between the Baotu spring level and precipitations of different stations (2012–2016).

| Precipitation Station | Time Scale | Distance (km) | 2012 | 2013 | 2014 | 2015 | 2016 | $\sum$ [a] | Rank |
|---|---|---|---|---|---|---|---|---|---|
| Donghongmiao | | 8.5 | 0.246 ** | 0.365 ** | 0.373 ** | 0.120 * | 0.300 ** | 1.404 | 1 |
| Jieshou | One day | 45.7 | 0.313 ** | 0.325 ** | 0.310 ** | 0.116 * | 0.267 ** | 1.331 | 2 |
| Qiujiazhuang | | 11 | 0.277 ** | 0.331 ** | 0.281 ** | 0.145 ** | 0.280 ** | 1.314 | 3 |
| Xinglong | | 8.7 | 0.251 ** | 0.363 ** | 0.250 ** | 0.124 ** | 0.262 ** | 1.25 | 4 |
| Donghongmiao | | 8.5 | 0.524 ** | 0.573 ** | 0.543 ** | 0.2 | 0.471 ** | 2.328 | 1 |
| Qiujiazhuang | Five days | 11 | 0.508 ** | 0.558 ** | 0.452 ** | 0.263 * | 0.453 ** | 2.234 | 2 |
| Jieshou | | 45.7 | 0.537 ** | 0.535 ** | 0.507 ** | 0.2 | 0.411 ** | 2.215 | 3 |
| Xinglong | | 8.7 | 0.496 ** | 0.594 ** | 0.419 ** | 0.235 * | 0.466 ** | 2.21 | 4 |
| Jiesou | | 45.7 | 0.625 ** | 0.556 ** | 0.557 ** | 0.3 | 0.567 ** | 2.557 | 1 |
| Donghongmiao | Ten days | 8.5 | 0.586 ** | 0.463 ** | 0.640 ** | 0.2 | 0.607 ** | 2.546 | 2 |
| Qiujiazhuang | | 11 | 0.567 ** | 0.459 ** | 0.521 ** | 0.334 * | 0.592 ** | 2.473 | 3 |
| Shihutong | | 43.5 | 0.499 ** | 0.520 ** | 0.568 ** | 0.3 | 0.549 ** | 2.422 | 4 |

Note: Only the precipitation stations whose comprehensive correlation coefficients rank the top 4 under each time scale are selected. ** The correlation passes the significant test at level 0.01 (double-tailed). * The correlation passes the significant test at level 0.05 (double-tailed). [a] The sum of correlation coefficients for each year.

In 2015, all stations showed lower coefficients than those of other years (Table 2), which is related to meteorological conditions and human activities [43]. Specifically, the decline in total precipitation and precipitation intensity and the increase in groundwater extraction are the main reasons for the weak correlation between the spring level and precipitation. Through ranking the comprehensive correlation coefficients, it can be obtained that the precipitation of Donghongmiao station, which is located in the direct recharge zone of exposed limestone area, has the greatest correlation with the Baotu spring level. Table 2 also demonstrates that the distance between springs and precipitation stations is not a significant factor that influences the correlativity. It is also related to the development degree of karstification near the gauge stations, the flow direction and hydraulic gradient of karst water, rainfall intensity and duration, and so on.

According to the above findings, the precipitation at Donghongmiao station is identified as having the greatest impact on the springs in the Jinan spring basin. One previous study made the notion that there is a strong association between precipitation of Xinglong station and the Baotu spring level [44]. Table 2 also shows that Xinglong station has a great correlation with spring level but ranks behind Donghongmiao station, and the two rainfall stations are within a short distance from each other in the direct recharge area of the spring basin. This discrepancy may be due to the differences in data and analysis methods. These

conclusions indicate the spatial heterogeneity of karst systems and can serve as a reference for the identification of preferential recharge zones in karst water.

### 4.4. Preferential Recharge Zones Constituted by Several Rainfall Stations

There is a strong correlation between the spring level and precipitation time series with a high intensity or long duration of precipitation, as discussed in the previous sections. As a result, this section selected the daily data of spring level (for both Baotu spring and Black Tiger spring) and precipitations from 16 stations in the wet seasons to perform the cross-correlation analysis method, which can make the lag response time scale accurate to days. Subsequently, the outcome of this process was used as the main basis for identifying preferential recharge zones in karst aquifers.

The interpolation result of comprehensive cross-correlation coefficients from 2012–2016 using the Kriging method in ArcGIS toolbox (Figure 8) demonstrates that the cross-correlation coefficient decreases to the southwest and southeast directions. This indicates that preferential recharge zones are possibly aligned from southwest to northeast and from southeast to northwest, in that the high cross-correlation coefficient means the precipitation around the site infiltrates the ground and flows to the discharge area relatively quickly. Furthermore, it is evident from the figure that the precipitations of Donghongmiao station, Xinglong station and Shaoer station belong to one group closely related to the spring dynamic, and these rainfall stations are all located in the direct recharge area. The distance between any two stations is less than 10 km and their corresponding lag days are comparable, indicating that the underground connectivity among these stations is stronger and groundwater flows faster within these zones.

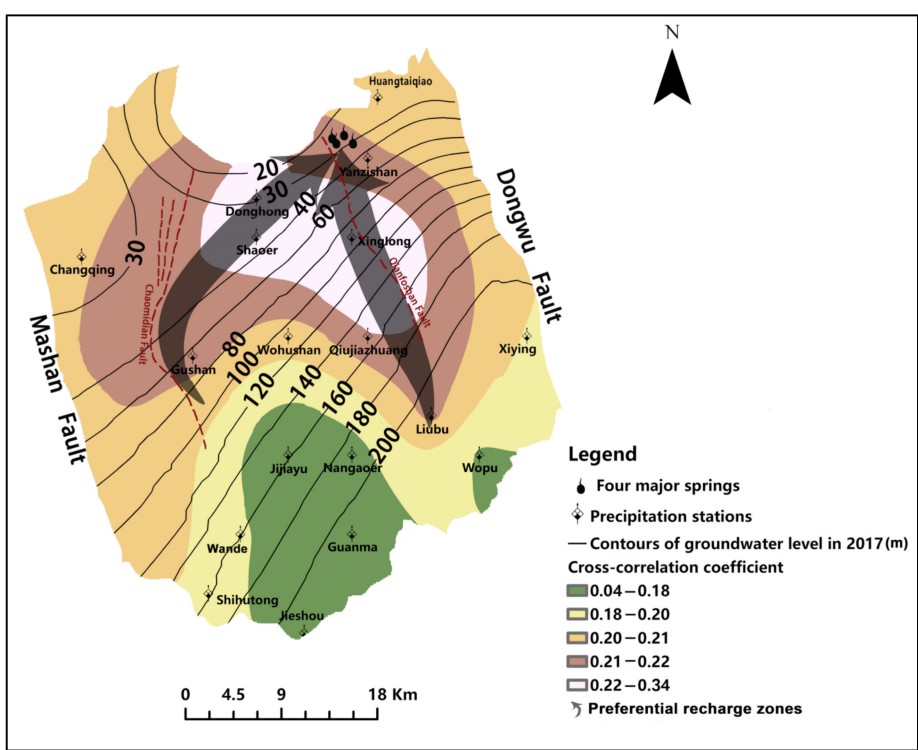

**Figure 8.** Distribution of preferential recharge zones identified according to Kriging interpolation of comprehensive cross-correlation coefficients and hydrogeological data.

The profile map (Figure 3) shows that the stratum system of the direct recharge area consists of upper, middle and lower Cambrian limestone, forming one typical karst water storage structure. The direct infiltration of precipitation further promotes the karst development between the Cambrian strata, which benefits the formation of preferential recharge zones [45]. Hence, there probably exists one southwest–northeast preferential recharge zone located in the vicinity of Donghongmiao station, Shaoer station and Gushan station,

ultimately pointing to the four major springs in the discharge area. The recharge zone approaches the Chaomidian Fault, where the karst aquifer system is well-developed with groups of fissures, conduits and fractures [46], facilitating the propagation of groundwater. It can be concluded that this preferential recharge zone is formed by the function of the stratigraphic contact zone and fault zone. Additionally, there is a northwest–southeast fault in the eastern part of the spring basin called Qianfoshan Fault, which is consistent with another attenuation direction of the cross-correlation coefficient, as shown in Figure 8. Therefore, the second preferential recharge zone is likely connected by Liubu station, Qiujiazhuang station, Xinglong station and Yanzishan station, extending along the fracture belt.

To verify the correlation between the spring level and these rainfall stations, multiple linear regression analysis was also carried out in SPSS. The results illustrate that the precipitations of Yanzishan station, Donghongmiao station, Qiujiazhuang station and Liubu station have greater impacts on the spring level, which means the precipitations from these stations can infiltrate the groundwater system quickly and easily, and then cause local groundwater levels to rise. Since the monitoring well density is low, the groundwater level contours can only display the overall flow direction of groundwater in the region, but cannot show the local flow field or the flow movement between two specific locations. A more accurate contour map of groundwater level can be drawn by increasing the distribution density of monitoring wells, and then the hydraulic connection between the rainfall stations and spring groups can be further verified through borehole data. To maintain the continuous flow of spring supply sources, the preferential recharge zones should be undamaged during the underground space development, and groundwater artificial recharge projects in the basin need to be carried out as close to preferential recharge zones as possible.

## 5. Discussion

A well-developed karstification zone, which is usually composed of fissures, conduits and caves, is ideal for the storage and movement of groundwater. Precipitation penetrates the karst aquifer and turns into karst water rapidly with the action of complex voids, which subsequently leads to a rise in the spring level. The results of correlation between the spring level and precipitation based on different statistical methods indicate that the precipitations of some gauge stations are truly more relevant to the spring level than those of other stations in the spring basin, which suggests that the precipitations of these stations recharge groundwater more effectively, implying that these areas have a high degree of karst development and can be recognized as preferential recharge zones. The rank of Pearson correlation coefficients (Table 2) shows that one factor influencing the correlation between the spring level and precipitation is the distance of the underground channel between two sites, which depends on the degree of karstification rather than the surface distance. The superposition analysis of the topographic map (Figure 1) and interpolation result (Figure 8) illustrates that terrains with lower slopes are more favorable for precipitation recharge, which is consistent with the findings of Andreo et al. [47]. In addition, Ma et al. [48] concluded that the Xinglong basin is a strong seepage zone by geophysical prospecting, exploration drilling and slug tests, and should be preserved for precipitation infiltration and groundwater propagation, which belongs to one part of the preferential recharge zones determined by this study.

In comparison to extensive geological surveys and borehole data integration, the identification of heterogeneity based on the correlation between the spring level and precipitation and hydrogeological data saves costs with high work efficiency, but also has uncertainty. The hydrologic process in which precipitation transforms into karst water by infiltration and emerges as springs by movement through heterogeneous karst systems is complicated, and it largely relies on the spatio-temporal distribution of precipitation, characteristics of the overburden layer and the development degree of karst. Radulovic et al. [49] used an index-overlay method for quantifying spatial variation of recharge, which took

into consideration eight factors affecting precipitation to recharge groundwater, including karstification, atmospheric conditions, lithology, overlying layers and so on. Therefore, it is a necessity to analyze the characteristics of the underlying surface and other factors that have great influences on precipitation infiltration before evaluating the correlation between groundwater dynamic and precipitation.

In the Jinan spring basin, urban development land is primarily concentrated in the discharge area, and so is the long-term groundwater extraction during the dry seasons. With the intention of conserving springs, the groundwater extraction is highly controlled, which only occurs when surface water supply is insufficient to fulfil people's subsistence and productivity. The objects of this study are the precipitation stations located in the direct and indirect recharge areas, which are almost evenly covered with woodland, grassland and farmland. The discrepancy of land use types around different gauge stations was not significant, so the effect of the overburden layer on the precipitation infiltration compared to precipitation in the wet seasons can be neglected. Consequently, we focused on the direct relationship between the groundwater dynamic and precipitation. A significant similarity of fluctuations between spring level and precipitation suggests that precipitation has little difficulty penetrating the ground and draining towards springs, which can serve as one basis to identify preferential recharge zones in karst regions.

Although the study of detecting preferential recharge zones relies on the synthesis of statistical analysis and limited hydrogeological data, and needs further verification by field investigations, such as the measurements of water temperature and conductivity, these analysis results have determined the approximate positions of preferential recharge zones in karst water, and help save the efforts of geophysical exploration and drilling activities, assessing the vulnerability of groundwater and constructing underground works.

This study did not explore in detail human activities such as groundwater extraction and land-use cover change, which have negligible impacts on the results. However, growing human activities associated with urban development may change the hydrogeological conditions of the Jinan spring basin in the future, including infiltration capacity, flow directions, and groundwater–surface water interaction. As a result, these issues should be fully considered, and more hydrogeological investigations should be conducted to assist in identifying the preferential recharge zones.

## 6. Conclusions

Time series techniques are powerful tools for the study of behaviors and attributes of karst systems. This research selected several widely used methods to study the correlation and causation between spring level and precipitation in the Jinan spring basin. (1) One factor that influences the correlation between the spring level and precipitation is the distance of the underground channel between two sites, which depends on the degree of karstification rather than the surface distance. (2) The precipitation at Donghongmiao station was found to contribute the most significantly to the springs because of its large correlation coefficient with the spring dynamic. (3) Two preferential recharge zones were identified in the Jinan spring basin according to the comprehensive analysis of correlation coefficients and geological exploration data. The results represent a reference value for planning and construction of spring protection and water supply engineering in the Jinan spring basin.

This study is a step towards enhancing the understanding of precipitation infiltration and propagation in karst systems, rendering hydrogeological exploration more targeted and efficient. The paper only analyzed spring level and precipitation data for the identification of preferential recharge zones. We recommend future works evaluate the correlation between the spring discharge and precipitation to make comparisons with findings of the study. Investigating the heterogeneity of spatial distribution in karst regions by other statistical methods such as the geographical detector is also desirable for future work.

**Author Contributions:** Conceptualization, Y.C. and L.S.; methodology, Y.C.; writing—original draft preparation, Y.C.; writing—review and editing, Y.C., L.S., P.A.O., Z.X. and T.Q.; supervision, L.S.; project administration, L.S., H.L. and G.L.; funding acquisition, L.S. All authors have read and agreed to the published version of the manuscript.

**Funding:** This study was funded by the Major Scientific and Technological Innovation Project of Shandong Province (CN) (no. 2019JZZY020105).

**Institutional Review Board Statement:** Not applicable.

**Informed Consent Statement:** Not applicable.

**Data Availability Statement:** All data and materials used in the manuscript are compliant with field standards and are available from the corresponding author upon reasonable request.

**Acknowledgments:** This work is based on a research project "Numerical Simulation of Groundwater Flow in Four-Dimensional Geological Environment Platform of Jinan Spring Basin", funded by Jinan Rail Transit Group Co., Ltd. We acknowledge the material support of Jinan Urban and Rural Water Authority and Shandong Provincial Bureau of Geology & Mineral Resources. We also thank the reviewers and editors for their feedback and constructive suggestions.

**Conflicts of Interest:** The authors have no relevant financial or non-financial interest to disclose.

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
