# Peer review of "Identification of Preferential Recharge Zones in Karst Systems Based on the Correlation between the Spring Level and Precipitation: A Case Study from Jinan Spring Basin"

_water, doi:10.3390/w13213048_

Round 1
Reviewer 1 Report
Karst is a heterogeneous hydro-geo-morphological system in which it is difficult to predict the processes, distribution of karst features and groundwater. From this point of view, any new attempt to better understand karst is very welcome. It is also important to collect new data from different parts of the world. For all these reasons, the paper is interesting. However, from the reviewer's point of view, it is necessary to rewrite and reorganise some parts of the paper, as well as to clarify some applied and discussed concepts.
I suggest to be stricter in applying IMRAD structure of the paper. In several paragraphs of Discussion methods are described (e.g. 313-317; 422, etc.).
Please improve the description of the study area. The article is published in an international journal aimed at international readers. These are not familiar with local geography and local conditions. It is also not reasonable to cite an article written in Chinese. All geographical names mentioned in the paper must be marked on the map. For the karst aquifer, it is important to understand the geology of the recharge area; at a minimum, create a general geologic map showing the major faults. Such a map is more important than a land use map for understanding the processes. It is also useful to illustrate the conditions on site at the springs with some photographs.
Human activity is mentioned in several places in the paper. Conditions are mentioned that affect the water balance, but I can not find any further information about how much water is extracted and under what conditions.
From a statistical point of view, the reviewers do not believe that it is necessary to present all the details about statistical methods, such as: linear trend analysis, correlation and cross-correlation analysis, Man-Kendall. These statistical tests and analyses are now so obvious that they are clear and known to every hydrologist. The reviewer suggests that the explanation of mutation point detection (lines 544 - 560) be moved to the main body of the paper. It would also be useful to explain in more detail the advantages of this statistical method and why it is used. Also provide original references to this method.
The use of time series methods in karst hydrogeology has a long tradition. Beginning with the French hydrogeologist Mangin, there is an extensive literature on the application of such methods in karst hydrogeology. I suggest citing some classic papers relevant to your study. However, randomly selected references that use similar methods are not helpful. A good place to start in citing and locating these references is the textbooks by Ford & Williams, and White.
The reviewer has trouble accepting the general methodology. Water level measurements are used in the analyses. And why? They are very often not representative because they depend on the morphology of the riverbed. Such analyses are usually done using discharge data. It is also very strange that two springs 1.5 km apart have almost the same water level. Please provide a rationale for using such a data set.
I cannot agree with the terminology "preferential runoff belts" or "groundwater runoff belts". To the reviewer's knowledge of the literature on karst hydrogeology, no such terminology exists. Nor is it correct in relation to the classical paradigm of karst hydrogeology. In karstified aquifers, there are zones of different degrees of karstification and different zones of different (tectonic) structure that affect the preferred pathways within the aquifer. Thus, geologically, there are no such belts. As can be seen from the application of statistical techniques in this paper, these are more likely to be artefacts resulting from the kriging process.
I disagree with the application of Thiessen (line 170) polygons for precipitation water balance. Today this method is not acceptable. Why not using kriging?
In line 183 is written: “Daily groundwater level values of confined aquifers were measured by 28 monitoring wells …” If there are confined aquifers given hydrogeological interpretation of the area are somehow vague. Please give more information on confined aquifers.
Strange is also terminology direct, indirect and discharge area of the spring. Please explain what do you mean with this? From where water balance in Table 1 is coming?
For which year Figure 7 is constructed?
I also suggest omitting general statements that apply in any karstic aquifer (e.g. 468-470). If such statements, known to any hydrologist dealing with karst, are excluded, the paper may be more effective.
The reviewer is not a native speaker and therefore not qualified to judge the language. His impression, however, is that the language can be improved. There are several sentences where the order of words needs to be changed. The reviewer recommends that the paper be proofread by a native speaker or someone with a good command of English. Please check how you use the adjective karst/karstic; harmonize it throughout the paper.
Reviewer 2 Report
The authors present an analysis of the relationship between precipitation and spring discharge for the Jinan spring basin in northern China. The authors describe the study area, some of the data and methods used, and conclude, through various forms of correlation analysis, that precipitation time series at certain stations have higher correlation with spring level than others, and that this correlation peaks at different lags depending on the year.
While the general topics of understanding the impact of meteorological forces on water supply and groundwater in general in karst systems are important ones, the information presented in this study does not synthesize data and analysis sufficiently to provide substantial insight on these topics. The current manuscript is well-suited as a technical note, but would need further elaboration and clarification before being acceptable as a research article. Some general notes/suggestions for improving the manuscript are listed below:
- The use of the term "preferential runoff belt" is somewhat confusing in this context. It seems perhaps that "preferential recharge zone" (as indicated in the legend of Figure 7) would be a more accurate terms.
- Analysis of correlation is helpful, but it doesn't necessarily explain conclusively that water flowing from one region to a discharge point happens in a particular way. Consider augmenting the existing precipitation-spring level analysis with other data or modeling - is there water temperature or conductivity data? does the current conceptualization of the system manifest as the same correlations in a computational model?
- The apparent trend in spring level and the lack of trend in precipitation data is an important aspect to consider, but is inadequately addressed in the current manuscript. Have groundwater extraction patterns or intensities changed? Has some aspect of precipitation intensity changed (i.e. less intense rainfall events that may recharge less)?
- The scope of the analysis is limited to a few years, which may obscure longer period oscillations (in climate, for example) that may be relevant to understanding trends. Consider expanding to include more data where possible.
- Why is there a shift in the dominant correlation at different lags between years? Why is the overall correlation in 2015 (among different stations) lower than in the other years? These questions are relevant to a broader (and perhaps more generalizable) understanding of hydroclimatic impacts on flow and discharge in karst systems and should be addressed in this manuscript.
- Where does the water budget information in Table 1 come from? Provide a citation for this, otherwise explain its development. Perhaps more importantly, further explanation of the ~20% discrepancy in the water budget is needed. This suggests an incomplete conceptualization of important water budget components.
- The conceptualization and interpretation of the groundwater system is somewhat unclear, especially as it pertains to the identification of recharge zones and the interpretation of groundwater flow direction. The manuscript references confined conditions in some portion of the system, but a direct recharge zone implies unconfined conditions. A more thorough and articulated hydrostratigraphic conceptual model, based on the available data, would be tremendously helpful conveying to the reader how and where the recharge and groundwater discharge are believed to occur and evolve. Within this framework, the authors can also confirm that the groundwater level data is consistently applied (all within the same hydrostratigraphic unit) to better support the groundwater contours and flow direction interpretation.
- If much of the groundwater extraction occurs in the discharge area, but that extraction is from a confined aquifer that supports the springs, then that extraction (or other development) may be having a disproportionate impact on spring conditions
- The arrows on Figure 7 identifying 'preferential recharge zones' are confusing - they appear to reflect a recharge-to-discharge flow path. This requires some clarification. If they are meant to reflect groundwater flow paths, then the orientation of the left arrow parallel to groundwater level contours seems inconsistent with principles of groundwater flow.
- Revisions to word usage and phrasing can help to more precisely convey the motivation for and findings from this study. Specific attention should be paid to how the correlation relationships are interpreted.
Reviewer 3 Report
No comments
Author Response
Thank the reviewer for the approval of this manuscript.
Round 2
Reviewer 1 Report
I would like to thank the authors who responded constructively to my comments and suggestions.
In Response 3: “We added some descriptions in the Discussion section to clarify (lines 533-534).” I don’t see much improvement in that part of the paper.
Despite the fact that there is a correlation between discharges and water levels in the spring area methodologically I cannot methodologically agree with this approach. A high correlation can only be explained by the fact that most measurements have a linear h=h(Q) relationship. In reality, such a relationship is never linear, especially at the extremes (low and high water); streambed profiles are often not uniform rectangles. Water level can only be a heuristic - indirect - indicator of hydrologic conditions in the karstic aquifer. The indication of discharges Q provides more information than the indication of water levels h. Based on the responses, it appears to the reviewer that the authors do not have enough information about Q. Therefore, I suggest that an explanation needs to be given as to why water levels h and not Q are given. If the authors think that it is useful to report and analyze the water levels h, you should also illustrate this with the correlation diagram h=h(Q) and present it for the data you have.
I agree with the statement, "... Thiessen polygon method is relatively convenient and simple, so we chose to use it.", but it is wrong; especially when you have relief differences in the catchment and spatial trends in precipitation. I do not understand why you did not use the Kriging method, because you used that interpolation method for other calculations. The application of this method cannot be that you have now read more literature and will use a different method in the future as you stated in your Response 8. Please provide a reasonable justification in your paper as to why you used the Thiessen polygon method.
Lines 170-171 “ … but the contour map cannot show the local flow field, which is complicated and volatile.” What do you mean with volatile? Strange terminology.
Comment on statement in Response 10 and lines 165-167 "Since the elevation of the discharge area is lower than that of the direct and indirect recharge area, ... " When we talk about gravity-induced flow, discharge area is ALWAYS lower than the recharge area. For hydrogeologic understanding of groundwater flow, the statement "... which contributes to the groundwater discharge from some surface outcrops." is also contradictory. Do you understand groundwater to mean only water that does not flow from karstic springs? Water that flows from springs is also groundwater. Please clarify the terminology.
Please explain somewhere what is mutation phenomenon in the time series.
This is only a comment from the point of view of the discussion. As the authors make clear in the literature, karstic/karst is not used consistently. The reviewer is not a native speaker and therefore not qualified to judge the use of English words, but he believes that the word karstic should be used in the function of an adjective and the word karst in the function of a noun.
Reviewer 2 Report
I thank the authors for the thorough consideration of comments and their effort in improving the manuscript. I suggest a few edits before recommending the manuscript for publication.
- Water budget (Table 1): Specify that the data in the table is for a single year (2012, as noted) and may not reflect a long-term steady-state condition. Similarly, it would be helpful to explicitly state that the authors attribute the positive discrepancy to an increase in aquifer storage.
- Conceptual hydrogeological model: In the response to my original comments, the authors explain the complexities of their conceptualization of the system. In particular, the confined-unconfined nature of the Cambrian-Ordovician karst system. I found this explanation helpful and I think it would improve the manuscript considerably if some of this explanation was included in Section 2.
- General comment: It is clear from the authors' response to comments that the intent of this manuscript is to explore only the relationship between precipitation and spring response, and to use that information to map out likely areas where precipitation has the most impact. It is important then to identify confounding factors (development, groundwater extraction, land cover change); the authors have provided a good foundation for this in the manuscript. A more explicit statement of these factors and how the authors hypothesize analysis of these factors in future work may affect the conclusions in this study would greatly improve the interpretability of the work done and reinforce and support the scope of this study.
